# Perception of the Neighborhood Environment, Physical Activity by Domain and Sitting Time in Brazilian Adults

**DOI:** 10.3390/ijerph192315744

**Published:** 2022-11-26

**Authors:** Diego Sales, Victor Matsudo, Mauro Fisberg, Clemens Drenowatz, Adilson Marques, Gerson Ferrari

**Affiliations:** 1Centro de Estudos do Laboratório de Aptidão Física de São Caetano do Sul (CELAFISCS), São Caetano do Sul 09521-160, Brazil; 2Departamento de Pediatria, Universidade Federal de São Paulo, São Paulo 04023-061, Brazil; 3Instituto Pensi, Fundação José Luiz Egydio Setubal, Hospital Infantil Sabará, São Paulo 01228-200, Brazil; 4Division of Sport, Physical Activity and Health, Pädagogische Hochschule Oberösterreich, 4020 Linz, Austria; 5CIPER, Faculdade de Motricidade Humana, Universidade de Lisboa, 1499-002 Lisbon, Portugal; 6ISAMB, Universidade de Lisboa, 1649-028 Lisbon, Portugal; 7Faculty of Health Sciences, Universidad Autónoma de Chile, Providencia 7500912, Chile

**Keywords:** built environment, physical activity, sedentary time, active transportation

## Abstract

This study aimed to examine the association between the neighborhood environment and domain-specific physical activity and sitting time in Brazilian adults. This cross-sectional study included 1803 adults (53.7% women) from Brazil’s five regions (North, Northeast, Midwest, Southeast and South). The perception of the environment was evaluated via a questionnaire. We considered seven indicators of the neighborhood environment: land use mix-diversity, land use mix-access, street connectivity, walking/cycling facilities, aesthetics, safety from traffic and safety from crime. Using the IPAQ, we evaluated physical activity (transport and leisure) and sitting time. Overall, land use mix-diversity (β = 3.22; 95% CI = 0.26, 6.19), land use mix-access (β = 2.27; 95% CI = 0.76, 3.79), and walking/cycling facilities (β = 2.42; 95% CI = 0.35, 4.49) were positively associated with leisure-time physical activity (min/week). On the other hand, only land use mix-diversity (β = 3.65; 95% CI = 0.63, 5.49) was positively associated with transport physical activity (min/week). No neighborhood environment indicator was associated with sitting time (min/day). Perception of the neighborhood environment was associated with physical activity (transport and leisure), while no significant associations occurred with sitting time among the five regions of Brazil. The neighborhood environment can be a viable component in the promotion of physical activity, but geographic diversity must be considered.

## 1. Introduction

International consensus shows that different movement behaviors that include sedentary time and physical activity (PA) have a crucial impact on health [1]. There is considerable data on the detrimental health effects of sedentary time and the health benefits of PA [1,2]. Specifically, sedentary behavior is associated with increased risks of cardiovascular disease and mortality [1,3,4], while PA is associated with a lower prevalence of cardiovascular diseases, depression, blood pressure, breast and colon cancer and mortality [5,6,7]. Therefore, the World Health Organization recommends that adults perform at least 150 to 300 min/week of moderate to vigorous PA, 75 to 150 min/week of vigorous PA, or an equivalent combination of both intensities. The same organization, however, does not establish any recommendation for sedentary time due to the different cutoff points used in previous studies [7].

Brazil has one of the largest territories in the world and a high social income inequality (the third most unequal country in the world) [8]. In addition, in recent years, Brazil has undergone socioeconomic, demographic and epidemiological changes due to accelerated urbanization, which has led to improvements in health indicators while decreasing the level of PA [9,10]. For instance, Brazil has a high prevalence of leisure-time physical inactivity (73%), with considerable differences between the main regions (North region with the lowest prevalence of inactive men [63%] compared to South [75%]) [11]. In addition, high air pollution, high social inequality, rising crime rates and population density are common characteristics of Brazil, which can contribute to high sitting time and physical inactivity [12,13,14]. In 2006, McNeill et al. provided a taxonomy of social environmental dimensions to bring clarity to the association between the neighborhood social environment and PA [15]. In 2019, Kepper et al. [16] updated a conceptual framework for how the neighborhood environment could affect PA and offered specific recommendations in order to guide future research and expand the literature. The authors concluded that nine dimensions of the neighborhood environment (for instance: crime and safety; economic and social disadvantage) were key influencers of PA [16].

Ecological models hypothesize that health behavior alterations are influenced by psychological, social, policy, and physical environmental features [17,18,19]. As a result, ecological models have now been proposed by researchers in the field. They assume that the built environment (land use mix-diversity and access, street connectivity, walking/cycling facilities, aesthetics, safety, among others), the natural environment (spaces in which there were no alterations made by man, such as beaches) and the social environment (sustenance providing through the incentive of other individuals or domestic animals to execute PA) have a strong relationship with the practice of PA. Environmental features therefore, are a critical component in intervention strategies [20,21,22,23,24]. Particularly the neighborhood environment, based in part from the urban planning literature [20,21,22], is believed to effect PA. The criterion validity and reliability of the neighborhood environment has been recognized in several countries across different continents [25,26,27,28]. For instance, test–retest reliability of the individual items (i.e., land use mix-diversity, residential density, land use mix-access, neighborhood aesthetics and neighborhood safety) of the Neighborhood Environment Walkability Scale-Abbreviated (NEWS-A) was moderate to high [27]. Significant associations were observed between the NEWS-A subscales [29] as well as self-report measures of PA [30,31]. Several reseachers and organizations have use environmental and policy interventions to encourage plans for PA promoting environmental modifications [32,33,34].

In recent decades, there has been increasing attention in investigating how factors in the built environment can impact PA levels and sitting time. For example, perceived environmental indicators such as residential density, safety, street characteristics were associated with walking [35,36,37,38]. Particularly residential density, crime and aesthetics were associated with high sedentary time [39,40]. In addition, a study conducted in eight Latin American countries found that neighborhood aesthetics and environmental safety were associated with increased PA [41]. Nevertheless, most of the studies in this line of research were carried out in the United States and Europe [42,43]. Given the socioeconomic, demographic and population differences across countries, these results, however, may not be transferable.

In Brazil, studies carried out in specific regions showed associations between residential density, public lighting and a positive perception of the neighborhood with PA and sedentary time [44,45,46,47,48]. Still, most of these studies did not investigate PA by domain in Brazil’s five main regions (North, Northeast, Midwest, Southeast, South). Brazil is politically and geographically divided with diverse physical, demographic, and socioeconomic characteristics. For instance, the North and the Northeast are the most deprived. The Southeast has greater economic power and includes more than 40% of the Brazilian inhabitants [49]. The South has the smallest territorial area, comprises 14% of the total population, and is one of the two wealthiest regions. The Center West has an intermediate stage of development that has accelerated since Brazil’s capital was moved to Brasilia in the 1960s. The concentration of poor socioeconomic situations in the North and Northeast regions and disparities in the distribution of health services and resources are reflected in a steep North–South gradient in health outcomes [49]. Given the great inequality and regional diversity in Brazil, comparisons between regions, generalizations regarding the impact of the environment on PA and sedentary time may be problematic [50]. Even though a study showed an association between availability of public space and greater chances of leisure-time PA in adults, the authors acknowledged differences in the magnitudes of these effects across the five main regions of Brazil (for instance: there were greater chances in the South region) [51].

Thus, examining how neighborhood environment indicators are associated with different domains (transport and leisure) of PA and sitting time in Brazil across its main geographic regions can be useful in guiding strategies and public policies that utilize the neighborhood environment for promoting PA in these regions. At this time only a few studies with a representative sample across the five regions of Brazil have been conducted [51,52,53]. Therefore, this study aimed to examine the association between the perception of the neighborhood environment and PA domains as well as sitting time in adults from the five main urban regions of Brazil.

## 2. Materials and Methods

### 2.1. Study Design and Sample

The Brazilian Nutrition and Health Study (Estudo Brasileiro de Nutricao e Saúde—EBANS) is a cross-sectional, representative, population-based study with participants aged 15–65 years. Data were collected between October 2014 and July 2015. EBANS is part of the Latin American Nutrition and Health Study (ELANS), from eight Latin American countries. Details about EBANS and ELANS have been previously published [54].

EBANS relied on population level data from the five main urban regions (North, Northeast, Midwest, Southeast and South) of Brazil with people from dissimilar socioeconomic levels (low, medium, and high) [55]. The study was approved by the Ethics Committee of the Federal University of Sao Paulo (number: 31670314.8.0000.5567). All participants received instructions and provided informed consent before data collection. All aspects of the study were conducted following the Declaration of Helsinki. For this article, we excluded adolescents aged 15–17 years and participants with incomplete or invalid data. Thus, the final sample consisted of 1803 participants (Figure 1).

### 2.2. Neighborhood Environment

To assess the characteristics of the perceived environment, the participants responded the NEWS-A translated into Portuguese for use in Brazil [56]. The validity and reliability of the NEWS-A has been established in previous studies within different contexts [30,57,58].

The characteristics of the built neighborhood environment were assessed based on the subsequent contexts: land use mix-diversity, land use mix-access, street connectivity, walking/cycling facilities, aesthetics, safety from traffic and safety from crime.

The scale of environmental use diversity was assessed by the perception of walking proximity to home for 23 distinct types of destinations, with responses ranging from 1–5 min of walking to >30 min of walking. The remaining six scales were estimated based on the average ratings of items answered on a four-point Likert scale (1 = strongly disagree to 4 = strongly agree). Answers were scored in a regular direction with increased mobility and safety receiving higher values and individual items were reversed when required.

### 2.3. Physical Activity and Sitting Time

Self-reported PA and sitting time were assessed using the International Physical Activity Questionnaire (IPAQ)—extended version according to the last seven days. The IPAQ has been validated using the CSA accelerometer to measure total PA in adults from diverse countries [59].

Participants were instructed on reporting frequency and duration of PA (≥10 min) for transport (walking and cycling) and leisure domains at dissimilar intensities (walking, moderate and vigorous). IPAQ data was converted to min/day and the results display min/week for each PA domain (transport and leisure). Details on the PA assessment using the IPAQ in the ELANS study have been previously published [60].

In addition, participants received instructions on how to estimate the amount of time (min/day) spent sitting at work, at home and at leisure for weekdays and weekends [61]. Overall daily average sitting time was calculated as weekday minutes × 5 + weekend day minutes × 2)/7.

### 2.4. Covariates

Sociodemographic data from participants, such as age, sex, marital status (single, married, widowed, and divorced), and race/ethnicity (white, black, mixed race), others [indigenous, gypsy, Asian]) was also collected via questionnaire. In addition, education status was determined as elementary school, middle/high school or higher education.

The socioeconomic level was determined via a questionnaire adapted for Brazil based on economic indicators and legislative requirements established in the country. Socioeconomic level data were classified into three levels (low, medium and high) based on Brazil’s national indices [62].

Dietary intake was assessed via two in-person 24 h dietary recall interviews using the automated multiple-pass method [63,64]. The food and beverage intakes were recorded using the Nutrition Data System for Research Software (NDS-R version 2013) [65]. Energy intake was further used as potential confounder to estimate the associations between perception of the neighborhood environment and PA as well as sitting time. Details are reported in more detail elsewhere [41,66].

Anthropometric measurements, including body weight and body height, were taken according to standardized procedures [54]. Body weight was measured to the nearest 0.1 kg using a portable scale (Seca Corporation, Hamburg, Germany) after all outer clothing, heavy pocket items, shoes, and socks were removed. Body height was measured to the nearest 0.5 cm without shoes using a Seca 213 portable stadiometer (Seca Corporation, Hamburg, Germany) with the participant’s head in the Frankfurt Plane [2,67]. Body mass index (kg/m^2^) was calculated (body weight [kg]/body height [m^2^]).

### 2.5. Statistical Analysis

The normality of data distribution was verified with Kolmogorov–Smirnov tests. We used mean, standard deviation (SD) and percentage to describe the data. Due to the nonparametric nature of PA and sitting time, we used the median and interquartile range.

Chi-square tests (categorical variables), One-Way analysis of variance (continuous variabels: age, energy intake, body mass index and perceived environment scores) and Kruskal–Wallis test (continuous variabels: PA domains [leisure and transportation] and sitting time) were carried out to compare the diferences between urban regions (North, Northeast, Midwest, Southeast and South) in the aforementioned variables.

We performed linear regression analysis (β coefficient and 95% confidence interval [95% CI]) to verify the associations between the perception of the neighborhood environment with the PA domains (leisure and transportation) and sitting time. Due to the non-normality of the data, minutes of each PA domain (transport and leisure) and sitting time were log-transformed for the linear models and the unstandardized coefficient values were back-transformed into min/week in order to increase the clinical value of the results. We adjusted the models in the analysis for age, sex, marital status, race/ethnicity, education level, socioeconomic level, energy intake, and body mass index. In addition, we considered *p* < 0.05 as a significant level and all analyzes executed using SPSS software version 26.

## 3. Results

Table 1 describes the characteristics of the participants for the total sample and by the different regions of Brazil. The total sample consisted of 1803 adults with a mean age of 37.9 (SD: 13.1) years, 53.7% female, 49.7% married, 41.6% race/ethnicity white and more than 45% of low or medium educational or socioeconomic level. The mean values of energy intake was 1837.5 (SD: 603.9) kcal/day. Regarding transport PA, participants reported an average of 141.5 (SD: 205.1) min/week with a difference of 104.9 (SD: 75.5) min/week between the North and Midwest regions. Regarding leisure-time PA, the mean was 125.3 (SD: 252.6) min/week with a difference of 123.6 (SD: 242.6) min/week between the same regions. The mean sitting time was 216.9 (SD: 152.6) min/week with a difference of 33.4 (SD: 8.4) min/week between the South and Midwest regions. There were significant differences between regions for age, marital status, race/ethnicity, energy intake, transportation PA, leisure PA and sitting time.

Regarding the description of the indicators of the perception of the neighborhood environment, the North and Midwest region had the highest and lowest scores, respectively for land use mix diversity. In the land use mix-access, the difference was greatest between the North and Center-West region. There were significant differences between regions for specific perceived environment scores (Table 2).

In addition, the North region had the highest perception of street connectivity, while the Southeast region had the lowest perception on this environmental component. Regarding walking/cycling facilities, the Midwest and South regions had the lowest and highest perception, respectively. For aesthetics, South and North showed differences of 0.3 points between these regions. In addition, the North region reported lower perceptions of safety from traffic compared to the South region. The scores for safety from crime showed a difference between the South and Northeast regions of 0.4 points for this variable (Table 2).

There was a positive association between land use mix-diversity and transport PA (β = 3.65; 95% CI = 0.63, 5.49) across the entire sample. However, when analyzing by specific regions, we found a negative association in the Northeast region (β = −6.73; 95% CI = −12.03, −1.16) and positive in the Southeast region (β = 4.23; 95% CI = 1.09, 7.73). In addition, significant associations occurred between street connectivity and transport PA in the North region (β = 10.67; 95% CI = 1.43, 19.91). Walking/cycling facilities were positively associated with transport PA only in the Midwest region (β = 0.53; 95% CI = 0.01, 1.06). Furthermore, transport PA was positively associated with aesthetics in the southern region (β = 5.57; 95% CI = 0.50, 10.63) (Table 3).

Overall, we observed positive associations between land use mix-diversity (β = 3.22; 95% CI = 0.26, 6.19), land use mix-access (β = 2.27; 95% CI = 0.76, 3.79), and walking/cycling facilities (β = 2.42; 95% CI = 0.35, 4.49) with leisure PA (min/week). Better perceptions of land use mix-access were associated with leisure-time PA in the South region (β = 10.37; 95% CI = 4.33, 16.41); while land use mix-diversity was associated with leisure PA in the Southeast region (β = 5.42; 95% CI = 1.80, 9.04). On the other hand, better street connectivity was associated with lower leisure-time PA in the Midwest region (β = −7.46; 95% CI = −14.66, −0.26) (Table 4).

For the total sample, we did not find a significant association between the different indicators of the neighborhood environment and sitting time. When analyzing regions, we found a direct association between land use mix-access and min/day of sitting time in the South region (β = 13.36; 95% CI = 2.15, 24.56). There was also an inverse relationship between land use mix-diversity and sitting time in the North region (β = −60.49; 95% CI = −99.91, −21.07) and Northeast (β = −11.39; 95% CI = −21.45, −1.33) while a direct association occurred in the Southeast region (β = 10.57; 95% CI = 2.57, 17.54). Perception of walking/cycling facilities (β = 17.72; 95% CI = 4.46, 30.98) and better perception of aesthetics (β = −14.06; 95% CI = −26.13, −1.99) were associated with a lower sitting time in the Midwest region (Table 5).

## 4. Discussion

The aim was to analyze the associations between the neighborhood environment and PA by domain as well as sitting time in Brazilian adults from the five main regions of Brazil. For the total sample, our results show that different indicators of the neighborhood environment were associated with leisure PA (i.e., land use mix-diversity, land use mix-access and walking/cycling facilities). At the same time, associations with transport PA were limited (i.e., only land use mix-diversity). Specifically, land use mix-diversity, land use mix-access, street connectivity and walking/cycling facilities were associated with leisure PA in different regions of Brazil. Further, land use mix-diversity, street connectivity and walking/cycling facilities were associated with transport PA in specific regions. No neighborhood environment indicator was associated with sitting time in the total sample. However, land use mix-diversity, street connectivity, walking/cycling facilities and aesthetics were associated with sitting time in specific regions of Brazil.

Considering the results by specific regions, a better perception of land use mix-diversity was associated with higher transport PA in the Southeast region. These results are in agreement with previous studies that showed positive associations between transport PA and perceived land use mix-diversity [35,41]. As in other studies carried out in the South and Southeast region of Brazil, it was shown that perception of a more suitable built environment for walking/cycling was associated with greater chances for active transport [68,69]. Establishments such as pharmacies, schools, banks, adv. located close to residences can motivate walking as a means of transport while increasing distances from these establishments can be a barrier to active transport. In addition, these results reinforce the benefits of reducing dependence on automobiles for such displacements and positively impacting health and the environment by reducing the emission of polluting gases produced by these motor vehicles. However, the Northeast region showed a decrease in PA for transport, even with the improvement in the perception of this environmental variable. This may be explained by an increase in the production and use of motorcycles as a means of transport in this region. The Northeast region also had the highest number of motorcycle sales in 2013 [70], as well as an increase in the perception of social advantage when using automobiles [71,72].

Our results further show an association of the built environment (land use mix-diversity, land use mix-access walking and cycling facilities) with leisure-time PA. When checking for specific regions, the Midwest region showed a decrease in leisure PA with a better perception of street connectivity. Factors such as the replacement of leisure-time PA with sedentary behaviors may explain these findings [73]. In the South region higher leisure PA was associated with a higher perception of land use mix-access, while in the Southeast region higher leisure PA was associated with a higher perception of land use mix-diversity. These results agree with previously published studies with the Latin American population, which showed positive associations between the perceived built environment and leisure-time PA in Brazil [41]. In addition, another study with adults from twelve countries indicated positive associations of residential density and land use mix-diversity with increased chances of leisure-time PA. In specific regions, better perceptions of access to public spaces were also associated with greater chances of leisure-time PA in all regions of Brazil [51]. Our findings show the importance of urban planning to improve infrastructure conditions (sidewalks and signs, for example) and safety for leisurely walking, as well as increasing access to public transport that facilitates access to parks and recreational areas. Further, more public spaces close to homes seem to be effective strategies for increasing PA in this domain.

A study by Florindo et al., with adult individuals from the city of São Paulo, showed that as the diversity of destinations near their homes increased, there was a decrease in sitting time [74]. In addition, street connectivity and aesthetics were associated with shorter sitting time, however, higher residential density was associated with longer sedentary time [39]. The present study also showed associations between the perceived built environment and sitting time for specific associations. Land use mix-diversity, land use mix-access, street connectivity and walking/cycling facilities were associated with higher sitting time in the North, Midwest, Southeast and South region. This relationship can be explained by a greater preference for screen-related behaviors on several leisure occasions [73].

The literature reports few interventions in Latin America. However, intervention policies in a built environment could favor transportation and leisurely walking among Brazilian adults, if adopted. For example, programs to encourage active transport in Colombia help promote the practice of total PA in which major avenues are closed to walking and cycling on certain days of the week. In addition, infrastructure changes to increase safety, such as providing bike lanes, increasing residential density, providing more green areas and parks, along with an increase in public transport such as “train” and “metro” close to homes in Sao Paulo seem to be good strategies through of the “Master Plan” objectives to increase transport and leisure PA [75,76,77,78]. In Brazil, the National Urban Development Policy is being formulated, which aims to propose an agenda with sustainable urban development objectives among the five regions of Brazil [79].

Given the cross-sectional design of the study causal inferences cannot be made as. It is also worth mentioning that there was no question as to where the individuals practiced PA. Thus, individuals who lived in neighborhoods whose environmental conditions were less conducive to the practice of PA could have practiced it elsewhere. The sample was composed to a larger extend of white people who lived in the Southeast region, which may affect generalizability to other ethnicities and Brazilian regions. In addition, the perceptions of the built environment and PA were assessed via questionnaires, which have a risk of response bias even though all questionnaires have been previously validated. There was also no in depth evaluation of climate and other specific characteristics of each region. Nevertheless, the present study has strengths. EBANS is a nationwide study with a representative sample of the five main regions of Brazil and analyzed individuals of all socioeconomic levels. We analyzed the associations of the built neighborhood environment with PA by domain in Brazil’s five main regions, allowing us to obtain a general parameter of the relationship between environment and PA in Brazil and its distinct characteristics between regions. The data from the present study can provide directions and strategies for public health policies, considering the different regional characteristics.

## 5. Conclusions

The current study showed that the various indicators of the neighborhood environment were more strongly associated with leisure-time PA than transport PA. Specifically, land use mix-diversity, land use mix-access, street connectivity and walking/cycling facilities were associated with leisure-time PA in different regions of Brazil. Different neighborhood environment indicators were also associated with sitting time in specific regions of Brazil but not the total sample. Given the consistently shown impact of the neighborhood environment on PA across diverse regions, this aspect should be considered more strongly in health promotion strategies. Environmental changes may also lead to more sustainable changes due to the lasting effect on the living situations compared to intervention strategies targeting behavioral changes directly.

## Figures and Tables

**Figure 1 ijerph-19-15744-f001:**
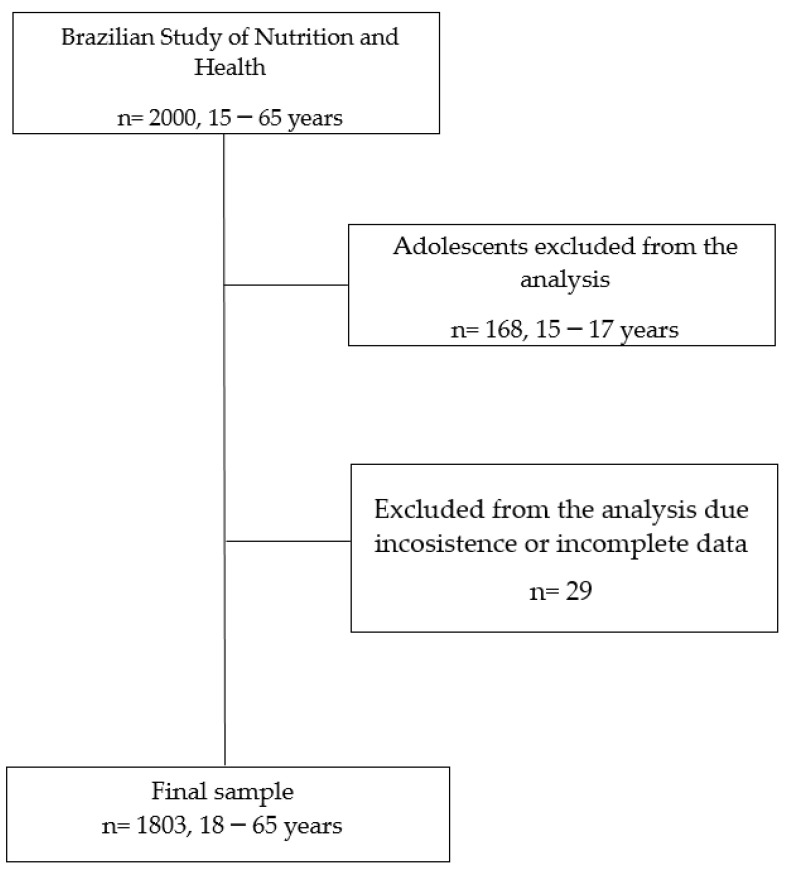
Flowchart for the selection of participants.

**Table 1 ijerph-19-15744-t001:** Sociodemographic characteristics, physical activity and sitting time of participants.

Variables	Overall	North	Northeast	Midwest	Southeast	South	*p*-Value
Simple size (*n*)	1.803	120	334	158	1.009	182	
Age (mean, SD)	37.9 (13.1)	36.5 (12.4)	37.3 (12.6)	36.1 (13.3)	38.1 (13.2)	40.0 (13.4)	0.038 ^1^
Sex (%)							
Men	46.6	52.5	41.0	43.7	47.5	47.3	0.220 ^2^
Women	53.7	47.5	59.0	56.3	52.5	52.7	
Marital status (%)							
Single	38.7	30.8	45.8	38.0	39.6	25.8	<0.001 ^2^
Married	49.7	64.2	47.0	50.0	48.0	54.4	
Widowed	3.6	1.7	1.2	6.3	3.6	7.1	
Divorced	8.0	3.3	6.0	5.7	8.8	12.6	
Race/ethnicity (%)							
White	41.6	16.5	19.4	41.3	48.7	57.6	<0.001 ^2^
Black	20.9	15.5	23.8	24.5	20.8	16.4	
Mixed	18.6	28.2	34.6	11.6	13.4	18.6	
Indigenous	2.4	13.6	1.9	5.2	0.8	2.8	
Other	16.6	26.2	20.4	17.4	16.3	4.5	
Education level (%)							
Low	45.6	49.2	52.1	35.4	44.7	45.1	0.089 ^2^
Medium	45.4	36.7	42.5	50.6	46.7	44.5	
High	9.0	14.2	5.4	13.9	8.6	10.4	
Socioeconomic level (%)							
Low	46.2	61.7	65.9	39.9	40.6	36.6	0.094 ^2^
Medium	45.2	28.3	31.7	36.7	51.5	53.3	
High	8.6	10.0	2.4	23.4	7.8	10.4	
Energy intake [kcal/day, mean (SD)]	1837.5 (603.9)	2077.3 (750.9)	1921.5 (610.3)	1784.5 (579.8)	1719.7 (565.9)	1952.0 (576.2)	<0.001 ^1^
Body mass index [kg/m^2^, mean (SD)]	27.0 (5.6)	27.2 (4.3)	27.2 (6.1)	27.1 (5.8)	26.8 (5.0)	26.2 (4.9)	<0.408 ^1^
Transportation PA [min/week, median (P25–P75)]	70.0 (0.0–175.0)	105.0 (40.0–280.0)	60.0 (0.0–140.0)	30.0 (0.0–121.2)	70.0 (0.0–160.0)	140.0 (60.0–225.0)	<0.001 ^3^
Leisure PA [min/week, median (P25–P75)]	0.0 (0.0–120.0)	2.5 (0.0–360.0)	0.0 (0.0–123.7)	0.0 (0.0–30.0)	0.0 (0.0–90.0)	0.0 (0.0–180.0)	<0.001 ^3^
Sitting time [min/week, median (P25–P75)]	180 (101.4–435.1)	201.4 (95.7–327.8)	182.1 (78.9–326.7)	158.5 (92.8–267.8)	180.0 (102.8–260.4)	201.4 (128.5–317.1)	<0.001 ^3^

SD: standard deviation; %: percentage; P25: percentile 25; P75: percentile 75; min/week: minutes per week; PA: physical activity. ^1^ One-Way analysis of variance. ^2^ Chi-square test. ^3^ Kruskal–Wallis test.

**Table 2 ijerph-19-15744-t002:** Overall and regions specific perceived environment scores.

Neighborhood Environment	Overall	North	Northeast	Midwest	Southeast	South	*p*-Value *
Land use mix-diversity ^1^	2.9 (0.3)	3.3 (0.3)	2.9 (0.3)	2.8 (0.2)	2.9 (0.4)	3.0 (0.4)	<0.001
Land use mix-access ^2^	2.5 (0.8)	2.9 (0.7)	2.8 (0.7)	1.9 (0.7)	2.5 (0.8)	2.8 (0.6)	<0.001
Street connectivity ^2^	2.7 (0.5)	3.1 (0.6)	2.7 (0.4)	2.8 (0.5)	2.6 (0.6)	2.8 (0.5)	<0.001
Walking/cycling facilities ^2^	2.3 (0.5)	2.3 (0.7)	2.3 (0.5)	2.1 (0.5)	2.3 (0.5)	2.4 (0.5)	0.001
Aesthetics ^2^	2.5 (0.7)	2.5 (0.8)	2.4 (0.5)	2.7 (0.6)	2.5 (0.8)	2.7 (0.5)	<0.001
Safety from traffic ^2^	2.4 (0.4)	2.2 (0.5)	2.3 (0.3)	2.4 (0.3)	2.4 (0.4)	2.3 (0.4)	0.001
Safety from crime ^2^	2.3 (0.5)	2.2 (0.6)	2.2 (0.4)	2.2 (0.4)	2.4 (0.6)	2.6 (0.4)	<0.001

Results presented as mean (standard deviation). ^1^ 5-points scale: ≤5 min: 1; 6 to 10 min: 2; 11 to 20 min: 3; 20 to 30 min: 4; >30 min: 5. ^2^ 4-points scale: strongly disagree: 1; disagree: 2: agree: 3; strongly agree: 4. * One-Way analysis of variance for comparison between urban region and perceived environment scores.

**Table 3 ijerph-19-15744-t003:** Association (β [95% CI]) between perceived neighborhood-built environment and transport physical activity (min/week).

Neighborhood Environment	Overall	North	Northeast	Midwest	Southeast	South
	β	β	β	β	β	β
	(95% CI)	(95% CI)	(95% CI)	(95% CI)	(95% CI)	(95% CI)
Land use mix-diversity ^1^	**3.65** (**0.63, 5.49**)	−9.15 (−23.2, 4.98)	**−6.73** (**−12.03, −1.16**)	7.81 (−4.02, 1.96)	**4.23** (**1.09, 7.73**)	2.80 (−4.52, 10.14)
Land use mix-access ^2^	0.92 (−0.31, 2.17)	−0.14 (−8.36, 8.08)	0.15 (−2.57, 2.88)	3.18 (−0.82, 7.20)	−0.0.7 (−1.70, 1.56)	2.51 (−1.84, 6.87)
Street connectivity ^2^	−0.19 (−1.83, 1.44)	**10.67** (**1.43, 19.91**)	0.06 (−4.54, 4.68)	−3.77 (−9.35, 1.79)	−0.94 (−3.02, 1.12)	−3.23 (−8.36, 1.89)
Walking/cycling facilities ^2^	−0.04 (−0.21, 0.12)	−0.68 (−0.14, 0.06)	−0.08 (−0.46, 0.30)	**0.53** (**0.01, 1.06**)	−1.05 (−3.25, 1.23)	2.13 (−3.96, 8.22)
Aesthetics ^2^	0.49 (−0.80, 1.79)	0.47 (−7.12, 8.06)	−1.10 (−4.53, 2.33)	−5.45 (−10.67, 0.73)	0.61 (−0.95, 2.18)	**5.57** (**0.50, 10.63**)
Safety from traffic ^2^	0.47 (−1.78, 2.73)	−0.28 (−11.86, 11.29)	3.65 (−1.60, 8.91)	−6.69 (−16.40, 3.01)	0.33 (−2.59, 3.26)	5.06 (−1.18, 11.32)
Safety from crime ^2^	1.61 (−0.09, 3.32)	−1.49 (−11.09, 8.10)	−0.45 (−4.68, 3.77)	−3.02 (−10.06, 4.01)	2.10 (0.02, 4.18)	0.93 (−6.13, 7.99)

Values in bold indicates statistical significance (*p* < 0.05). β: regression coefficient; CI: confidence interval; min/week: minutes per week. Linear regression model adjusted for age, sex, marital status, race/ethnicity, education level, socioeconomic level, energy intake, and body mass index. ^1^ 5-points scale: ≤5 min: 1; 6 to 10 min: 2; 11 to 20 min: 3; 20 to 30 min: 4; >30 min: 5. ^2^ 4-points scale: strongly disagree: 1; disagree: 2: agree: 3; strongly agree: 4.

**Table 4 ijerph-19-15744-t004:** Association (β [95% CI]) between perceived neighborhood-built environment and leisure physical activity (min/week).

Neighborhood Environment	Overall	North	Northeast	Midwest	Southeast	South
	β	β	β	β	β	β
	(95% CI)	(95% CI)	(95% CI)	(95% CI)	(95% CI)	(95% CI)
Land use mix-diversity ^1^	**3.22** (**0.26, 6.19**)	−15.03 (−33.16, 3.10)	−4.90 (−11.74, 1.92)	−5.80 (−21.27, 9.67)	**5.42** (**1.80, 9.04**)	−2.47 (−12.94, 8.00)
Land use mix-access ^2^	**2.27** (**0.76, 3.79**)	−2.41 (−13.01, 8.18)	−2.03 (−5.36, 1.29)	2.41 (−2.84, 7.66)	1.88 (−0.01, 3.76)	**10.37 ** (**4.33, 16.41**)
Street connectivity ^2^	−0.51 (−2.51, 1.48)	3.12 (−9.09, 15.35)	1.31 (−4.31, 6.94)	**−7.46** (**−14.66, −0.26**)	−0.73 (−3.12, 1.65)	−4.21 (−11.53, 3.10)
Walking/cycling facilities ^2^	**2.42** (**0.35, 4.49**)	−2.90 (−12.67, 6.86)	4.16 (−0.49, 8.83)	8.09 (1.31, 14.86)	1.28 (−1.30, 3.87)	6.12 (−2.25, 14.78)
Aesthetics ^2^	0.06 (−1.51, 1.64)	−3.76 (−13.52, 6.00)	0.06 (−4.13, 4.25)	−5.59 (−11.77, 0.59)	0.80 (−1.00, 2.61)	0.17 (−7.14, 7.49)
Safety from traffic ^2^	−2.56 (−5.31, 0.18)	−2.95 (−17.87, 11.96)	−1.30 (−7.74, 5.13)	2.56 (−10.15, 15.28)	−2.92 (−6.29, 0.44)	2.98 (−5.98, 11.95)
Safety from crime ^2^	1.06 (−1.01, 3.15)	1.90 (−10.46, 14.28)	0.89 (−4.61, 6.06)	2.93 (−6.24, 12.11)	0.67 (−1.72, 3.07)	−1.53 (−11.61, 8.54)

Values in bold indicates statistical significance (*p* < 0.05). β: regression coefficient; CI: confidence interval; min/week: minutes per week. Linear regression model adjusted for age, sex, marital status, race/ethnicity, education level, socioeconomic level, energy intake, and body mass index. ^1^ 5 -points scale: ≤5 min: 1; 6 to 10 min: 2; 11 to 20 min: 3; 20 to 30 min: 4; >30 min: 5. ^2^ 4-points scale: strongly disagree: 1; disagree: 2: agree: 3; strongly agree: 4.

**Table 5 ijerph-19-15744-t005:** Association (β [95% CI]) between perceived neighborhood-built environment and sitting time (min/day).

Neighborhood Environment	Overall	North	Northeast	Midwest	Southeast	South
	β	β	β	β	β	β
	(95% CI)	(95% CI)	(95% CI)	(95% CI)	(95% CI)	(95% CI)
Land use mix-diversity ^1^	5.07 (−0.81, 10.96)	**−60.49** (**−99.91, −21.07**)	**−11.39** (**−21.45, −1.33**)	−2.32 (−32.81, 28.15)	**10.57** (**2.57, 17.54**)	−0.62 (−19.74, 18.48)
Land use mix-access ^2^	2.35 (−0.65, 5.35)	−13.19 (−36.86, 10.48)	−2.49 (−7.41, 2.43)	8.99 (−1.26, 19.25)	0.83 (−3.05, 4.72)	**13.36** (**2.15, 24.56**)
Street connectivity ^2^	−0.36 (−4.33, 3.60)	**28.26** (**1.39, 55.14**)	1.71 (−6.06, 10.03)	−11.74 (−25.98, 2.48)	−2.97 (−7.90, 196)	−7.35 (−20.71, 5.99)
Walking/cycling facilities ^2^	3.55 (−0.55, 7.65)	−5.65 (−27.61, 16.29)	4.35 (−2.56, 11.27)	**17.72** (**4.46, 30.98**)	0.28 (−5.06, 5.63)	13.30 (−2.44, 29.06)
Aesthetics ^2^	0.26 (−2.86, 3.40)	−8.21 (−30.14, 13.72)	−1.32 (−7.52, 4.87)	**−14.06** (**−26.13, −1.99**)	1.72 (−2.01, 5.47)	8.26 (−5.02, 21.55)
Safety from traffic ^2^	−3.32 (−8.78, 2.35)	−8.21 (−30.14, 13.72)	−1.32 (−7.52, 4.87)	−7.25 (−40.76, 26.24)	1.72 (−2.01, 5.47)	8.26 (−5.02, 21.55)
Safety from crime ^2^	2.99 (−1.13, 7.13)	−7.51 (−35.27, 20.25)	1.96 (−5.66, 9.58)	−3.23 (−21.30, 14.82)	3.48 (−1.47, 8.44)	5.75 (−12.62, 24.12)

Values in bold indicates statistical significance (*p* < 0.05). β: regression coefficient; CI: confidence interval; min/week: minutes per week. Linear regression model adjusted for age, sex, marital status, race/ethnicity, education level, socioeconomic level, energy intake, and body mass index. ^1^ 5-points scale: 5 min: 1; 6–10 min: 2; 11–20 min: 3; 20–30 min: 4; >30 min: 5. ^2^ 4-points scale: strongly disagree: 1; disagree: 2: agree: 3; strongly agree: 4.

## Data Availability

The datasets generated and/or analyzed during the current study are not publicly available due the terms of consent/assent to which the participants agreed but are available from the corresponding author on reasonable request. Contact the corresponding author too discuss availability of data and materials.

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
