# Peer review of "Perception of the Neighborhood Environment, Physical Activity by Domain and Sitting Time in Brazilian Adults"

_ijerph, 2022, doi:10.3390/ijerph192315744_

Round 1

Reviewer 1 Report

This study seeks to examine the impact of the perception of neighbourhood environment on physical activity among Brazilian adults. They employed a questionnaire survey in 5 regions of Brazil. While the data showed non-normal distribution, the authors employed SPSS to examine the associations between the main variables. In reviewing the manuscript, a number of issues come to mind and are detailed below:

1.      Introduction: At first, in introduction, you must clearly state the need for this study (with a range of citations) and what has been solved in the prior studies, and what knowledge gap remains nonetheless. Therefore, the importance of the research, the theoretical gaps in the research and the significance of the research are not clearly explained.

2.      Theoretical framework is missing here. What are the underlying theories in this study? There are many other environmental and socio-demographic factors that might be related to PA. What are the justifications to select the following variables: land use mix-diversity, land use mix-access, street connectivity, walking/cycling facilities, aesthetics, safety from traffic and safety from crime?

3.      Literature: The authors need to provide more than a cursory and outdated view of the literature

4.      Analysis: Why the authors did not employ SmartPLS as this software does not require non-normal distributions? There are no comparisons among the five regions in terms of the associations between the variables. For example, the authors need to conduct an ANOVA test to see if there are any differences between the main variables. AND region plays a moderator role in research design (based on authors’ explanations). Where are the results of moderation analysis? One cannot simply conclude that the associations between main variables differ among regions.

5.      Discussion and conclusion: The discussion presented in this section is quite similar to the previous literature. I would like to see what the strength/uniqueness of this study is. Sometimes, studies have innovations in terms of research methods or data analysis ….. I don’t see any novelty in this research (at least the way that the authors developed their arguments).

6.      I ran Turnitin for this manuscript and the results show a very high percentage of similarities (31%). Please check.

Reviewer 2 Report

Your paper is well organised. The results and limitations of the study are clearly interpreted. 

45-47 Therefore, the World Health Organization recommends that adults perform at 45 least 150 to 300 minutes of moderate to vigorous PA, 75 to 150 minutes of vigorous PA, or 46 an equivalent combination of the both intensities. What is the time unit (per day, week?) Should be mentioned.

Reviewer 3 Report

Physical activity and sedentary behavior have a crucial impact on people's metabolic, cardiovascular, cancer and mortality health. This study aimed to examine the association between neighborhood environment perception and PA domains and sitting time in adults from the five main urban regions of Brazil.

The following comments are sent to the authors for their consideration:

1. Were diseases with cardiovascular diseases or diseases where a high percentage of physical activities is not recommended evaluated? Could this affect the results?

2. Were the questionnaires on physical activity and sedentary time applied by a surveyor or were they self-administered?

3. The physical activity and sitting time data had a non-parametric distribution, so why did you perform a linear regression model? Perhaps if you were able to use quantile regression models.

4. Why did they not adjust the data for the nutritional status of the participants? This variable has to do with the level of physical activity and sedentary time.

5. It is suggested that comparisons be made in Table 1 to assess whether there is a difference between the regions.

6. What is the reason that means (SD) and medians (IQR) of the same variable (physical activity and sedentary time) are reported in Table 1? In the statistical analysis section it is mentioned that the variables of physical activity and sitting time were non-parametric.

7. It is recommended that comparisons (p values) between the regions be obtained in Table 2, since it is not identified if there really is a difference between them or not.

8. To carry out a regression analysis, it is suggested before carrying out a bivariate analysis between the variables of interest, describing said statistical tests in the statistical analysis section.

Round 2

Reviewer 1 Report

I thank the authors for taking some of the recommendations on board. Overall, I am happy with some of the implemented changes. Below I will outline my concerns. 

Referring to my second comment on the original version of the manuscript, the authors did not include a theoretical framework and did not explain how they develop the study model. They added some information on the main problem in the Introduction, but there is no information on the theoretical/conceptual framework.

Referring to my previous comments (No 4), I would suggest the authors provide more justifications for the methods used to analyse the data.

And referring to my previous comments (No 5), the authors have only made minor changes to the conclusions. I would like to see the novelty and uniqueness of the study, what are the practical and theoretical implications of the study as the study was conducted in a less tested area.

I re-ran Turnitin for the revised manuscript and the results still show a very high percentage of similarities (32%).  

Author Response

Prof. Dr. Paul B. Tchounwou

Editor-in-Chief - International Journal of Environmental Research and Public Health

Thank you for considering our manuscript entitled "Perception of the neighborhood environment, physical activity by domain and sitting time in Brazilian adults”, which has been sent for review at the International Journal of Environmental Research and Public Health.

We have read the reviewer’s comments and addressed each of them carefully. These changes are highlighted in red in the manuscript. A copy with tracked changes has been submitted to the journal. 

Overall, we feel that revisions made have resulted in a stronger manuscript and that the findings represent a significant contribution to the literature on associations between neighborhood environment, physical activity by domain and sitting time. As suggested by the reviewers, we made the appropriate changes to have a more concise article with a better scientific structure. We trust that you will feel the same and we look forward to your editorial decision.

Yours sincerely,

Dr. Gerson Ferrari

Corresponding author (on behalf of the authors)

We would like to thank the reviewers for taking the time to review our manuscript and we appreciate their constructive and thoughtful comments. Based on these suggestions, we have revised our manuscript accordingly. Please find below our point-by-point responses to the news comments and suggestions.

Reviewer: 1

  1. Referring to my second comment on the original version of the manuscript, the authors did not include a theoretical framework and did not explain how they develop the study model. They added some information on the main problem in the Introduction, but there is no information on the theoretical/conceptual framework.

Author’s response: We have made appropriate changes in the introduction.

  1. Referring to my previous comments (No 4), I would suggest the authors provide more justifications for the methods used to analyse the data.

Author’s response: Many statistical methods require assumptions to be made about the format of the data to be analysed. The Kolmogorov-Smirnov test is 1 well-known and historically widely applied quantitative methods to assess for data normality. Kolmogorov-Smirnov tests compare the scores in the study sample with a normally distributed set of scores with the same mean and standard deviation; their null  hypothesis is that sample distribution is normal. Therefore, if the test is significant (P< .05), the sample data distribution is non-normal. The Kolmogorov-Smirnov test is more appropriate for big sample sizes (N > 50), as is the case of our study.

https://pubmed.ncbi.nlm.nih.gov/28787341/

https://www.academia.edu/40433614/Intuitive_Biostatistics_A_Nonmathematical_Guide_to_Statistical_Thinking_4th_Edition_by_Motulsky_1_1_

In this study, normality tests were conducted on continuous variables and indicated that the minutes of physical activity and sitting time data were not normally distributed. The Kolmogorov–Smirnov test is used to decide if a sample comes from a population with a specific distribution. Previous studies of physical activity and sitting time have also used the same statistical analyses.

https://pubmed.ncbi.nlm.nih.gov/17040259/

https://www.mdpi.com/1660-4601/18/9/4654

https://pubmed.ncbi.nlm.nih.gov/12493072/

https://pubmed.ncbi.nlm.nih.gov/17909412/

https://pubmed.ncbi.nlm.nih.gov/30293524/

We made comparison analyses between the regions using Chi-square tests, One-Way analysis of variance and Kruskal–Wallis test. We used the Chi-Square test because is a statistical procedure used to examine the differences between categorical variables.

Numerical data (i.e., perceived environment scores) that are normally distributed can be analyzed with parametric tests, that is, tests which are based on the parameters that define a normal distribution curve. In our study we used One-Way analysis of variance (parametric data) and Kruskal-Wallis test as the nonparametric equivalent of One-Way analysis. The One-Way analysis of variance and Kruskal-Wallis test are used to determine whether three or more independent groups (in our case, regions) are the same or different on some variable of interest (physical activity, sitting time and perceived environment scores).

https://pubmed.ncbi.nlm.nih.gov/27293244/

https://pubmed.ncbi.nlm.nih.gov/9413454/

https://pubmed.ncbi.nlm.nih.gov/28182832/

https://pubmed.ncbi.nlm.nih.gov/28295394/

  1. And referring to my previous comments (No 5), the authors have only made minor changes to the conclusions. I would like to see the novelty and uniqueness of the study, what are the practical and theoretical implications of the study as the study was conducted in a less tested area.

Author’s response:  Thank you for your suggestion. We have emphasized the consistency in the results across diverse regions in Brazil which provides additional support for the role of the neighborhood environment in health promotion. Such changes may also result in more sustainable lifestyle changes compared to behavioral interventions such as exercise programs.

  1. I re-ran Turnitin for the revised manuscript and the results still show a very high percentage of similarities (32%).

Author’s response: Thank you for your attention. The entire article has been reviewed and we have tried to decrease the percentage.

Reviewer 3 Report

The authors responded and made the suggested changes to improve the manuscript.

Author Response

Prof. Dr. Paul B. Tchounwou

Editor-in-Chief - International Journal of Environmental Research and Public Health

Thank you for considering our manuscript entitled "Perception of the neighborhood environment, physical activity by domain and sitting time in Brazilian adults”, which has been sent for review at the International Journal of Environmental Research and Public Health.

We have read the reviewer’s comments and addressed each of them carefully. These changes are highlighted in red in the manuscript. A copy with tracked changes has been submitted to the journal. 

Overall, we feel that revisions made have resulted in a stronger manuscript and that the findings represent a significant contribution to the literature on associations between neighborhood environment, physical activity by domain and sitting time. As suggested by the reviewers, we made the appropriate changes to have a more concise article with a better scientific structure. We trust that you will feel the same and we look forward to your editorial decision.

Yours sincerely,

Dr. Gerson Ferrari

Corresponding author (on behalf of the authors)

We would like to thank the reviewers for taking the time to review our manuscript and we appreciate their constructive and thoughtful comments. Based on these suggestions, we have revised our manuscript accordingly. Please find below our point-by-point responses to the comments and suggestions.

Reviewer: 2

  1. The authors responded and made the suggested changes to improve the manuscript.

Author’s response:  Thank you for your attention and suggestions. We believe that with your suggestions, we will have a better quality paper.
